# Female Genital Schistosomiasis (FGS) in Cameroon: A formative epidemiological and socioeconomic investigation in eleven rural fishing communities

**Makia Christine Masong**[1]*, **Godlove Bunda Wepnje**[2], **Ntsinda Tchoffo Marlene**[1], **Victoria Gamba**[3], **Marie-Therese Mengue**[1], **Estelle Kouokam**[1], **J. Russell Stothard**[4], **Albert Legrand Same Ekobo**[5]

**1** Department of Social Sciences and Management, Catholic University of Central Africa, Yaoundé, Cameroon, **2** Department of Zoology and Animal Physiology, University of Buea, Buea, Cameroon, **3** Department of Obstetrics and Gynecology, University of Nairobi, Nairobi, Kenya, **4** Department of Tropical Disease Biology, Liverpool School of Tropical Medicine, Liverpool, United Kingdom, **5** Department of Medicine and Biomedical Sciences, University Teaching Hospital, Yaoundé, Cameroon

* masongbye@yahoo.com

**Data Availability Statement:** All related data for this paper has been put in our manuscript and Supporting Information.

## Abstract

### Background

Female Genital Schistosomiasis (FGS) is most often caused by presence of *Schistosoma haematobium* eggs lodged in the female reproductive tract which results in chronic fibrosis and scarring. In Cameroon, despite high community prevalences of urine-patent *S. haematobium* infections, FGS has yet to be studied in depth. To shed light on the clinical prevalence and socioeconomic effects of FGS, we undertook a formative community-based epidemiological and qualitative survey.

### Method

A cross sectional multidisciplinary study of 304 girls and women from 11 remote rural fishing communities in Cameroon was undertaken using parasitological sampling, clinical colposcopy, and interviews. The lived experiences of those with FGS were documented using a process of ethnography with participant observation and in-depth interviews.

### Result

Amongst 304 women and girls aged >5 years (Median age: 18; Interquartile range: 9.6–28), 198 females were eligible for FGS testing and 58 adult women were examined by clinical colposcopy. Of these, 34 were positive for FGS (proportion: 58.6%; 95% CI: 45.8–70.4), younger girls showing a higher FGS prevalence, and older women not shedding eggs showing a pattern for cervical lesions from earlier infection. In a subset of women with FGS selected purposively (12/58), in-depth interviews with participant observation revealed out-of-pocket expenditures of up to 500USD related health spending for repeated diagnosis and treatment of gynecological illnesses, and 9 hours daily lost reproductive labour.

**Funding:** This study is funded through a fellowship offered by "l'Organisation de Coordination pour la lutte contre les Endémies en Afrique Centrale (OCEAC)", financed by the Ministry for Economic Cooperation and Development (BMZ) of the Federal Republic of Germany, through the KfW (German Developement Bank). "Projet Lutte Contre Les Maladies Tropicales Negligées en Afrique Centrale" (MTN) BMZ-Nr 2015.69.227 BMZ 2016.68.797. The funders had no role in study design, data collection and analysis, decision to publish, or preparation of the manuscript.

**Competing interests:** All authors declare no competing interests exist.

Psychosocial unrest, loss in social capital, and despair were linked with sub-fertility and persistent vaginal itch.

## Conclusion

With our first formative evidence on prevalence, socioeconomic effects and experiences of FGS amongst women and girls in Cameroon, we have clarified to a new level of detail the deficit in provision of and access to peripheral health services in remote areas of Cameroon. Using this information, there is now strong evidence for national programs and services on women's health and schistosomiasis to update and revise policies targeted on prevention and management of FGS. We therefore stress the need for regular provision of Praziquantel treatment to adolescent girls and women in *S. haematobium* endemic areas, alongside better access to tailored diagnostic services that can detect FGS and appropriately triage care at primary health level.

## Introduction

Neglected Tropical Diseases (NTDs), included under Target 3.3 of the Sustainable Development Goal (SDG) framework [1, 2] cause tremendous physical and socio- economic losses [3, 4] mostly seen with a reduced ability to work [5], from both physical debilitating and detrimental mental health effects [5–7]. Furthermore, out-of-pocket health related expenditures, as a result of misdiagnosis or lack of access to prompt diagnosis [8], can cause further suffering. Although most NTDs are not fatal [9], affected persons and their families can incur catastrophic health expenditure, and become less economically productive.

With regard to urogenital schistosomiasis, this disease can damage reproductive organs and tracts of both genders [10, 11]. Over the past decade, there is heightening awareness of Female Genital Schistosomiasis (FGS), in terms of diagnostic research [12–17], knowledge sharing [18–21] and pragmatic policy guides and information [22, 23], even though only a very small fraction of those with suspected FGS have been examined clinically [22].

Indeed this gap in routine detection of FGS within current primary health care settings means most endemic communities are still unaware of the existence or diagnosis of this important gender-specific manifestation of urogenital schistosomiasis [24–27]. The need to readdress this unfortunate oversight is important as there is unequivocal evidence of the clear link between FGS, and raised acquisition of Human Immunodeficiency Virus (HIV) [28–30] and cervical cancer [22]. The Central Africa region is a known host for the *Schistosoma haematobium* blood flukes, with high endemicity in several countries like Gabon [31], Democratic Republic of Congo [32], Central African Republic and Cameroon [33]. In Cameroon, though discussions have been held towards its management within the National program [34], the extent of FGS still has to be quantified, yet urogenital schistosomiasis has been formally reported back to 1949 [35], with several regions still hyper-endemic for *Schistosoma haematobium* [33]. Current estimates from the national control programme for schistosomiasis estimate more than 5 million people at risk of urogenital schistosomiasis [35, 36]. Intervention program for schistosomiasis in Cameroon is school based [37], though community based treatment is carried out in some highly endemic areas. Notwithstanding, treatment coverage is still incomplete and certain demographic groups are missed out or not met [37, 38], which is seen as one of the main challenges in preventive chemotherapy [39]. By shedding light on the

epidemiological state and lived experiences of girls and women infected with FGS in Cameroon, our work sets to unveil the local importance of this disease, giving particular leeway into its psychosocial and economic effects, further expounding on the importance of prevention, accessible diagnostics and drug availability for FGS and reproductive health care as a whole.

## Methods

Using case study sites for a context specific example, our study employed a cross discipline or mixed methods approach, using standard epidemiological and qualitative methods of investigation.

### Ethical considerations

The Cameroon National Ethics committee on Human Health Research (Ref N˚ 2020/07/1266/ CE/CNERSH/SP) approved the study. Administrative authorizations were gotten from the Regional Delegation of Public Health for the West Region of Cameroon (Ref N˚ 679/L/MIN-SANTE/SG/DRSPO/CBF), the District Medical Officer for the Malanteoun Health District (Ref N˚ 078/L/MINSANTE/DRSPO/DSMLT), and the Magba sub-Divisional Office (Ref N˚ 01/AR/F32.05/BAAJP).

All Participants gave written and verbal informed consent to participate. Participants, less than 18 years gave assent while their parents, husbands, or guardian gave consent for them to participate. In addition, oral consent was obtained from husbands or partners of married women before they were approached as required by the cultural regulations.

### Study site and participants

This study was carried out in the Matta Health Area, one of the 18 health areas found within the Malanteoun Health District in the West Region of Cameroon. The health district has an estimated total population of 125,564 people with 36,692 aged 5–14, and 88, 872 above 14 or below 5 years [40]. Mass drug administration with Praziquantel through school based intervention amongst 5–14 years is the method of control for schistosomiasis here (as everywhere else in Cameroon) [37] and as of 2019, 120,152 children were treated [41], with adults unaccounted for or unregistered till date. For decades now, the Malanteoun health district has been known to house hotspot areas with high transmission after treatment for schistosomiasis in Cameroon, amongst which Matta health area is the most significant [42]. Looking at its prevalence statistics for over a period of 10 years, the Matta health area has maintained a prevalence of more than 41% of continuous high transmission and endemicity since its first recording on school based prevalence in 1985 [43]. Our case study communities are remote water enclosed communities with little or no road access (mostly accessible only by boat) and are classical peasant in agriculture. Farming and fishing contributes more than 95% of the economic activities in this area, with the extra 5% involved in petit trading but still farm and/or fish as side or main economic activity (account from our ethnographic field diaries).

The Matta health area has one Government Integrated Health Centre and two private health centers [33], providing healthcare services to a general population of about 5000 persons. Spread out over 15 fishing Islands or Camps (locally known as 'Campoments'), two mainland villages, with a host of neighboring communities which though administratively are found within the Adamawa Region, are covered by the Matta health area, dependent on it for healthcare services (See Fig 1).

Matta was of specific interest for this study due to the presence of a water barrage constructed in the early 1960s as a diversion for the Mape Dam (3,300 million $m^3$), functioning since 1988 to preserve water and supply to the near regions. This barrage for over decades has

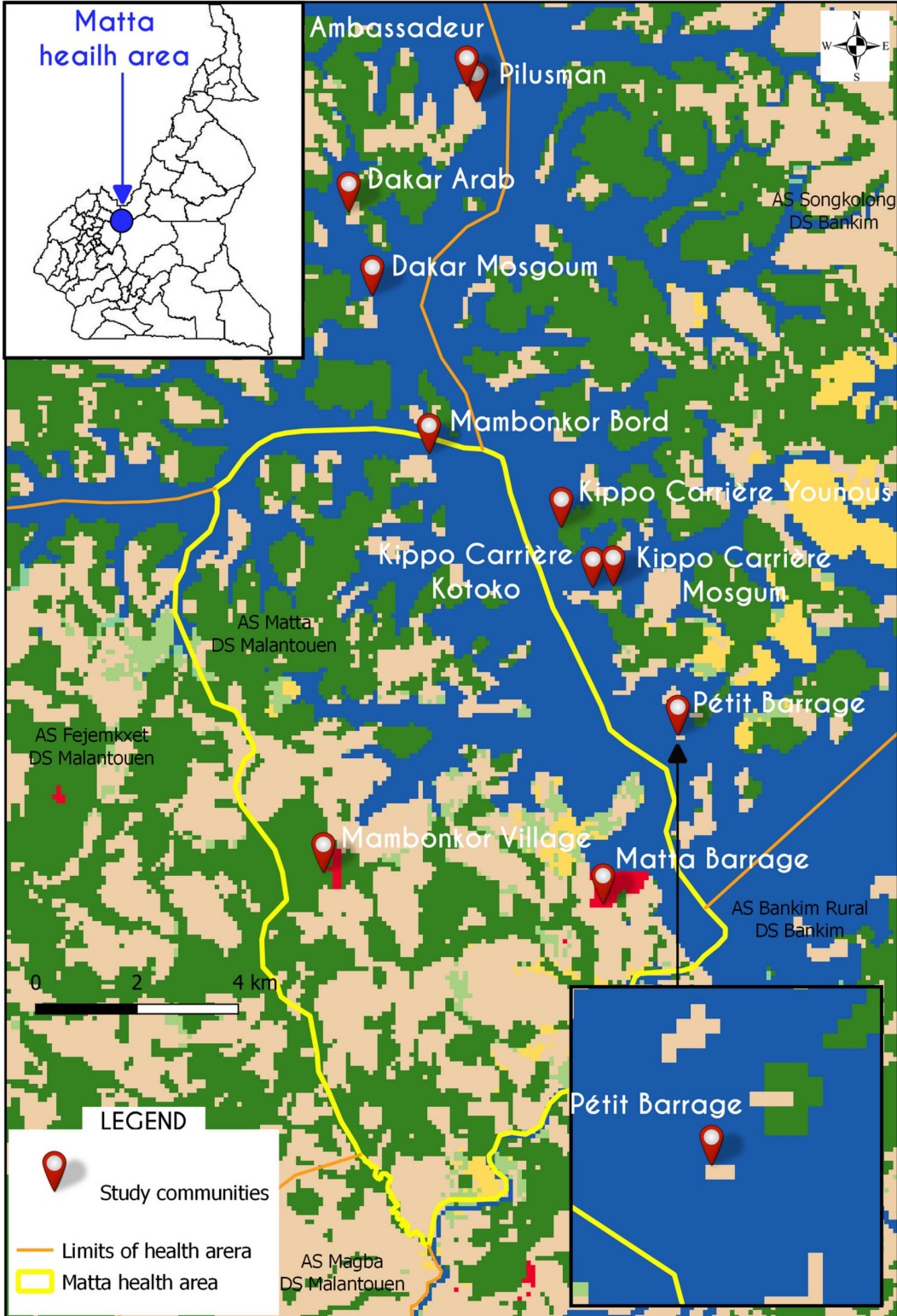

**Fig 1. Study sites within and around the Matta health area covered by the health district of Malanteoun.** Cameroon map showing the location of the Matta health area. Excerpt of Petit Barrage community, an example to show the remoteness and water enclosed nature of communities, only accessible through boats (fishing boats). Source data: Map base layer is from the Esri 2020 Land Cover map (https://arcg.is/0Kemuy). This dataset is available under a Creative Commons BY-4.0 license and any copy of or work based on this dataset requires the following attribution: This dataset is based on the dataset produced for the Dynamic World Project by National Geographic Society in partnership with Google and the World Resources Institute. Health boundaries from the Ministry of Public Health Cameroon (https://www.minsante.cm/); GPS coordinates collected during field survey.

**Table 1. Study site/population description (complemented with data from Malanteoun health district census, 2018–2020. S1 Fig).**

| Island | Total population size | Economic activity | Ethnic group | Number of women/girls (estimate) |
|---|---|---|---|---|
| Mambonkor Bord | 235 | Fishing | Massa, Arab, Mosgum, Massa, Njorku | 180 |
| Kippo Carriere (Kippo Kotoko, Kippo Mosgum, Kippo Younous) | 340 | Fishing (and light Farming) | Kotoko, Mosgum | 225 |
| Dakar (Dakar Arab and Dakar Mosgum) | 415 | Fishing | Arab, Mosgum | 300 |
| Petit Barrage | 220 | Fishing | Njorku, Akum | 115 |
| Ambassadeur (1) | 436 | Fishing and farming | Malian, Tikar, Arab | 260 |
| Pilusman | 143 | Fishing | Njorku | 80 |
| **Mainland** | | | | |
| Mambonkor Village | > 1000 | Farming | Tikar | >500 |
| Matta Barrage Village | >2000 | Farming (and light fishing) | Tikar, Kotoko, Mosgum, Arab | >1000 |

grown to attract a large population for mainly fishing activities, who subsequently settled around the barrage on little Islands–'Campoments (de peche)' or fishing camps (Fig 1). However, over the years, settlers migrated from different local ethnic groups mostly from the Far North (Massa, Kotoko, Mosgum, Arab), North West and West (mostly the Tikars, who are the indigenes of this area) regions of Cameroon, and different countries (Mali and Nigeria) to practice fishing for economic gain. Extensive fishing came to be (and is still) practiced on these Islands, with light farming (mostly the Tikars), adding to the fishing culture. Nevertheless, central to our study, are the fishing Islands which depend totally on the barrage for their source of water for drinking and household chores, and produces the sole source of income generation–fishing. In addition, 2 communities with a pure farming culture, located some distance away from the barrage (Mambonkor Village and Matta Village) were considered in our study to offer an alternate perspective or contrast.

Thus, for this study, we selected fishing Islands along the Mape Dam: Kippo Carriere (Mosgum and Kotoko), Dakar (Arab and Mosgum), Petit Barrage, Mambonkor Bord, Pilusman, Ambassadeur (Fig 1), and two mainland villages far from the lake. These mainland villages (Mambonkor Village and Matta Village) due to their distance from the barrage have some access to other water sources and with different socioeconomic culture (mostly farmers and petit traders). All case study sites were selected based on their presence within the Matta health area, endemicity of *Schistosoma haematobium* infection, proximity to the lake, socioeconomic culture (strictly fishing or fishing and/or farming), population size, and diversity in access to source of water (see Table 1). All communities selected for the study were georeferenced using a hand held GPS.

## Sample size determination

We determined our sample size based on the population estimate of our case study sites. The total population estimate of the study site was 5,000 people [40], where women represented 51% or 2,550 of the population, with age group 15–64 years representing 54.6% or 1,400 of the total population [40]. Using a simple random sampling technique, on the base of attaining a precision rate of 95% with an error margin of 5%, our sample size was statistically estimated using the population proportion formula as follows:

n = N*X / (X + N – 1),

Where,

X = Zα/2 ¬ *p*(1-p) / MOE2, and Zα/2 is the critical value of the Normal distribution at α/2 (for the confidence level of 95%, α is 0.05 and the critical value is 1.96), MOE is the margin of error (5%), p is the sample proportion (55%), and N is the population size (1400). It should be noted that the Finite Population Correction has been applied to the sample size formula.

Thus,

n = 1400*3.8 / (3.8 + 1400–1),

n = 383

While our target population was 383, logistic constraints meant 80% target recruitment was reached.

## Recruitment, testing and questioning

In total, 304 participants were initially recruited for urine collection, partly through voluntary response (for children and older women) and others (married girls between 14 and 25) by snowball sampling. Participants were contacted within the community through the traditional birth attendants or traditional midwives, and community drug distributors or health workers. We met participants at their homes or in meeting rooms allocated per Island by the village head. In a context where more than 90% of the women deliver at home, the traditional birth attendants are the reference points for women's reproductive health. They are very much respected and were most resourceful in our acceptance and trust by the study participants.

**a. Study design and questionnaire survey.** After urine collection from girls 5 years and above and women from individual houses, codes were given out to each participant and asked to present the code the following day when the test results were returned, at a specific meeting point preselected by each village/Island head (always a male). Identification papers were collected and double checked with our registered records, parasitological results were shared and a single dose of Praziquantel (40mg/kg) [23] was offered and administered to those who shed *Schistosoma* eggs in urine (from syringe filtration technique) and microhaematuria from dipstick tests. At this stage, women were selected for gynecological testing and administered questionnaires based on: age groups (14–17; 18–24; 25–35; 36 and above); urine test results (women found positive for urogenital schistosomiasis), self-narrated medical history or symptoms from responses to administered questionnaires (S1 Text) during urine sample collection. Trained medical personnel carried out gynecological examinations by visual colposcopy on consenting women or girls above 14, who had had a previous sexual encounter. These were women or girls who were either positive for urogenital schistosomiasis (UGS) in urine or were negative for UGS, but had self-reported FGS related symptoms indicative of underlying pathological signs of FGS such as cervical scaring and yellow sandy patches, amongst others. These images collected from the colposcopic reading were analyzed by trained personnel and cross-referred to the WHO FGS pocket atlas for similarities. Images collected were grouped by case, anonymized by giving a code per image, which was similar to pre-existing code from urine collection, and registered on questionnaires.

Of the 304 women and girls tested for microhaematuria and *S. haematobium* eggs in urine, 114 (37.5%) were either below 13 years of age, pre menarche, and/or virgins. For these young girls who could not undergo visual examination by colposcopy or answer to FGS related questions, few questions were asked to document some FGS related symptoms, in the presence of or in collaboration with their parent or guardian. Only urine was collected for microscopic examination to determine *S. haematobium* egg excretion, from which we infer a possibility for the presence of FGS in correlation with self-reported symptoms and infection amongst related household members or neighbors collected during our study.

For convenience, female participants were divided into three groups. The first group were girls ≤ 13 years of age who were diagnosed for *S. haematobium* infection through urine

microscopy, and positive cases were treated accordingly. The level of egg intensity in urine was quantified and communicated as possible FGS cases, though these were not confirmed by visual colposcopy. No questions related to symptoms of FGS was asked in this group, neither were they invited for interviews nor focused group discussions.

The second group were unmarried girls/virgins of post-menarche. The girls in this group as well, submitted urine samples for determination of *S. haematobium* infection, and were interviewed on FGS related symptoms. One Focus group discussion (FGD) was carried out here, involving a female health attendant at the health facility. This was to probe their knowledge on the different symptoms of FGS and the different local names given for these symptoms, and if present their therapeutic routes used. These girls as well, when found positive for UGS were treated with a single dose of 40mg/kg Praziquantel. The level of egg density in urine was quantified and communicated as possible FGS cases, though not confirmed visually.

Lastly, the third group were married girls and women/non-virgins. Following diagnosis of collected urine for determination of *S. haematobium infection*, a cervical examination was performed where an image of their cervix was taken to observe whether sandy patches and scaring from *S. haematobium* eggs on this surface occurred.

**b. Sample collection and processing.** On the day of enrollment, each participant was given a sterile, wide mouthed, screw capped plastic bottle carrying their identification information. About 250ml urine was collected from each participant. All samples were stored in cooling boxes containing cooler ice packs to prevent the eggs from hatching during transportation to the Health Centre laboratory. In the laboratory, urine was observed visually for macrohaematuria then tested for microhaematuria and proteinuria using urine reagent strips (Siemens Multistix 10). The urine samples were processed using the membrane filtration technique and examined microscopically for the presence of *S. haematobium* infection based on morphology of the ova. In brief, 10 ml of urine was filtered through membrane filter (Sterlitech Polycarbonate (PCTE) membrane filters, USA), the filter was removed, placed on a glass slide and stained with 1% Lugol's iodine solution [44, 45]. The slide was then examined using the Binocular Compound light microscope. Terminal-spined eggs, characteristics of *S. haematobium* were identified and counted manually. The egg load was defined by the number of eggs per 10ml of urine, categorized as light (<50 eggs/10 ml of urine) or heavy (≥50 eggs/10ml of urine) infection as defined by the WHO [46].

During colposcopy examination, a hand-held colposcope (EVA COLPO from Mobile ODT) was used and images were taken, then put into a coded database, which was reviewed by independent experts for evidence of the disease. A case was determined FGS positive if after visual analysis; sandy patches on homogenous yellow area, grainy sandy patches and abnormal blood vessels were found (according to the WHO FGS pocket atlas [23]).

**c. Close-ended structured questionnaires.** To women and post-menarcheal girls (who have had their first sexual intercourse) selected for the study, we asked questions (S1 Text) in a gender and culturally sensitive manner about the presence of typical symptoms of FGS such as vaginal discharge, contact bleeding, infertility or sub-fertility. In addition, questions about previous history of urogenital schistosomiasis or history of sexually transmitted infections (STIs) that did not respond to treatment and irregular and/or painful menstruation were asked.

To community leaders or local opinion leaders and women or girls who are living or have lived in/or near schistosomiasis-endemic areas, we questioned to identify communities or community members who do not have access to or benefit from mass drug administration, water source and water contact history (S1 Text). This included (parents or caretakers of) school-aged children who are not in school, adults at risk, and marginalized individuals (e.g. people living with disabilities, migrants, indigenous groups, etc.), in order to gather knowledge

on environmental practices, the equity gaps within these communities and knowledge of how to obtain Praziquantel for schistosomiasis and FGS prevention.

**d. Anthropological survey.**   After sample collection and processing, directly linked to the results, a total of 12 positive girls and women were purposively selected based on their age group, marital status, socio-cultural and physiological determinants such as: number of children, infertility or sub-fertility, menstrual irregularities, and vaginal discharges for in-depth interviews. In addition, participant observation with the construction of field diaries was used to enrich our knowledge and assist us in forming and confirming pre-selected hypothesis throughout the period of this research.

*i. In-depth Interviews and Focus Group Discussions.* After consenting, with the use of interview guides (S1 Text), single in-depth interviews with life history narratives were carried out with 12 women/girls. This was to probe on the lived experiences, perceptions and practices of post-menarche girls and sexually active women (also other community members) around the existing symptoms of FGS, its diagnosis, treatment and effects. Apart from the girls and women found positive for FGS, the "other community members" whom we have mentioned here include: community leaders; formal and informal health workers and some identified people within the community. In total four focused group discussions were carried out made up of one with married women > 30 years, one with younger (married) women ≥15, one with health workers, one with Massa women of all age groups.

*ii. Participant Observation/Ethnography.* Participant observation, as part of the process of ethnography was used in the quest of an emic or internal understanding of cultures and people. This was guided by ethnographic interviews, with in-depth description and study of our case study site contexts helping to situate our interviews and language gotten in the context of the discourses, aiding the understanding and consciousness of social life in our case study sites.

## Data analysis

All numerical data were double-entered using Microsoft Excel 2013 and checked, with tabulations, graphs and statistical analyses conducted using IBM Statistical Package for Social Sciences (IBM SPSS Statistics for Windows, Version 22.0. Armonk, NY: IBM Corp) and Graph pad prism version 8. Chi-square and Fisher's exact tests were used to check for associations between UGS, FGS, site, risk factors and morbidity. Univariable analysis and multivariable logistic regression analysis was conducted for *S. haematobium* associations with independent variables. Age group (young girls; adolescents; adults), village site, and economic activity were included as core variables in all regression models, and a stepwise regression approach was used to arrive at the most parsimonious adjusted model, applying a 5% level of statistical significance.

Using an inductive approach, at the end of each day of data collection, interview guides and interviews from recorded audios were relooked and main themes adapted, with an initial coding theme developed, which was updated from the beginning to the end of data collection with emerging themes. This strategy allowed new emerging themes to be identified and incorporated into the interview guides for subsequent interviews that ensured rich textured data quality and continued feedback. Interviews were recorded in Fulbe, English, French and pidgin-English, and the audios transcribed directly (for English or French audios) or translated from Fulbe, Kotoko, Massa and Arab into English before transcription. The team then developed an initial thematic framework that was used to code manually the transcripts and all materials including field notes. All data was organized by themes with some cross checking for consensus and quality. Once all data had been coded, similarities and differences within each code were reviewed to develop thematic charts with consideration of characteristics such as age, sex, marital status, residence, ethnical origin and number of children. Once the charts were

completed, the data was discussed at length by the research team to interpret descriptive and explanatory accounts of each emergent theme [47].

## Results

### A. Clinical and epidemiological variables

**i. Demographic characteristics of participants.**   The characteristics of participants is shown in Table 2. A total of 304 females from nine (9) Islands and two (2) Mainland communities were enrolled into the study. They were made up of girls and women aged 5 to 89 years old (Median age: 18; Interquartile range: 9.6–28), with married (88%) and unmarried women (12%), most of them (88.2%) living in a proximity of ≤100m to the lake.

After urine analysis for *S. haematobium* egg excretion, and colposcopic visual inspection of the cervix for characteristic lesions, the prevalence of FGS (with or without eggs in urine) was 58.6% from all selected sites. Images from colposcopy were cross verified with the WHO FGS pocket Atlas as reference, and a positive diagnosis was confirmed on the presence of at least two of these three main parameters: grainy sandy patch (56.9%); homogenous yellow sandy patch (55.2%); abnormal blood vessels 60.3%).

**ii. Parasitological and clinical description.**   In this study, 194 girls and women (63.8%; 95% CI: 58.3–69) were positive for UGS. Among the infected participants, 94 had light infection (1–50 eggs /10ml of urine) while 50 were heavily infected (≥50 eggs/10ml of urine). Of the 304 participants, a total of 198 females were eligible for FGS with 58 examined visually by colposcopy. The prevalence of FGS was 58.6%, 34/58 (95% CI: 45.8–70.4) for homogenous

**Table 2. Participant characteristics.**

| Variable | Category | %(n) |
|---|---|---|
| **Area of residence** | Ambassadeur | 8.9(27) |
| | Carrier Younous | 2.6(8) |
| | Dakar Arab | 6.6(20) |
| | Dakar Mosgum | 7.2(22) |
| | Kippo Carrier Kotoko | 19.4(59) |
| | Kippo Carrier Mosgum | 10.9(33) |
| | Mambonkor Bord | 14.5(44) |
| | Mambonkor Village | 8.6(26) |
| | Matta Barrage | 3.6(11) |
| | Petit Barrage | 10.9(33) |
| | Pilusman | 6.9(21) |
| **Age group** | Young girl (< 13 years) | 33.9(103) |
| | Adolescent (13–19 years) | 23.0(70) |
| | Adults (> 19 years) | 43.1(131) |
| **Marital status** | Single | 12.0(23) |
| | Married | 88.0(169) |
| **Level of education** | No formal education | 87.2(265) |
| | Atleast primary level | 12.8(39) |
| **Type of economic activity** | Fishing | 34.2((104) |
| | Both fishing/farming | 34.2(104) |
| | Farming | 9.9(30) |
| | No activity | 21.7(66) |
| **Proximity to lake** | ≤100m | 88.2(268) |
| | >100m | 11.8(36) |

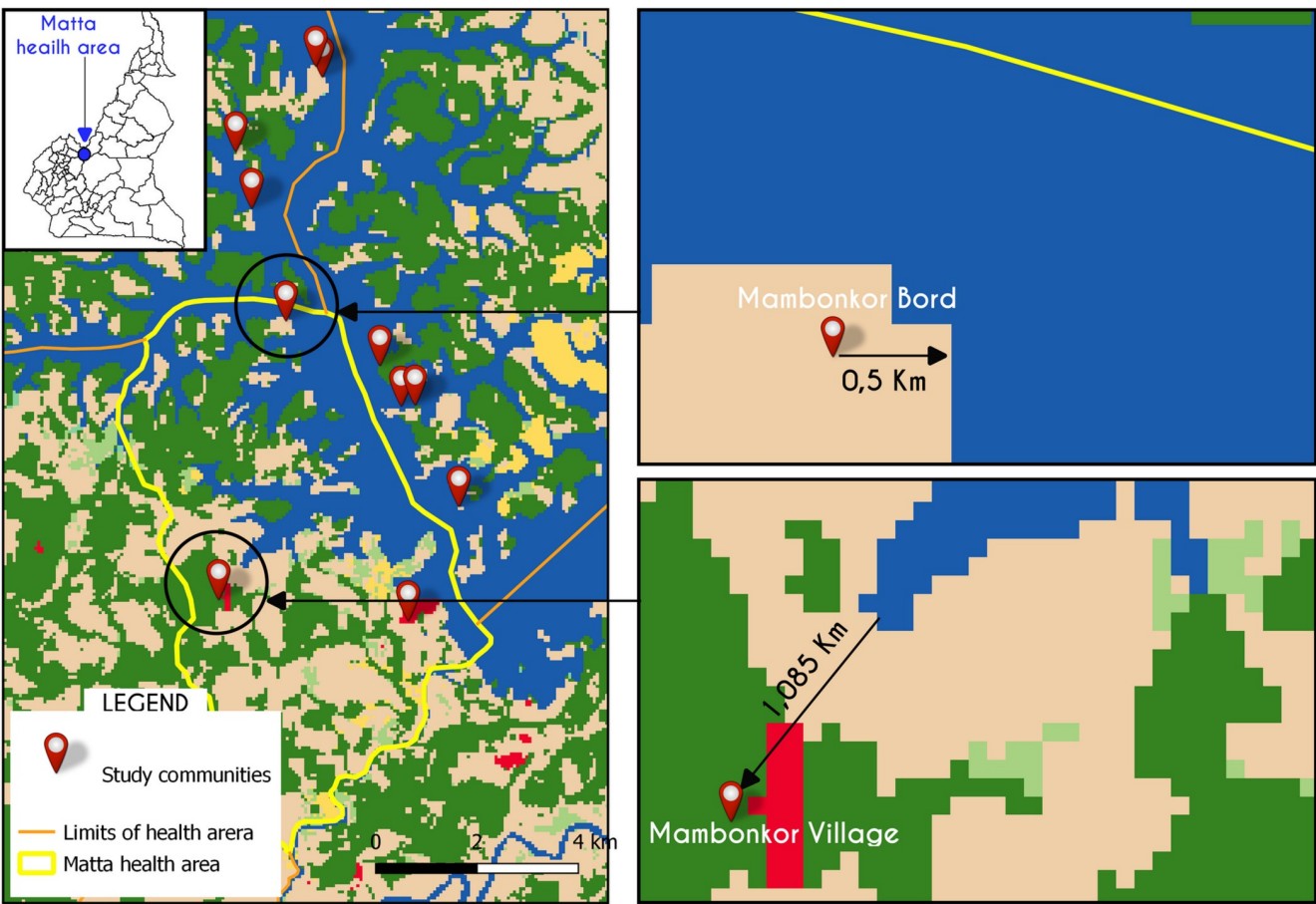

**Fig 2. Map showing different lake proximity of two study communities and highlights the FGS infection risks related to lake proximity.** A: Studied communities B: Mambonkor Bord C: Mambonkor Village. Source data: Map base layer is from the Esri 2020 Land Cover map (https://arcg.is/0Kemuy). This dataset is available under a Creative Commons BY-4.0 license and any copy of or work based on this dataset requires the following attribution: This dataset is based on the dataset produced for the Dynamic World Project by National Geographic Society in partnership with Google and the World Resources Institute. Health boundaries from the Ministry of Public Health Cameroon (https://www.minsante.cm/); GPS coordinates collected during field survey.

yellow sandy patches, grainy sandy patches and abnormal blood vessels/vascular changes. Apart from the 58.6% positive FGS, the remaining were either negative or had uncertain results (41.4%) (Where, apart from symptomatic questioning and urogenital schistosomiasis testing, a visual analysis by colposcopy was not carried out).

**iii. FGS and socio-demographic factors.** Differences in prevalence rate were considered along five types of variables: egg shedding in urine (urogenital schistosomiasis), socioeconomic culture, lake proximity, age, marital status, infertility and sub-fertility. Of these, association was reported only between proximity to lake ($\chi^2 2$: 8.676; df: 1; p-value = 0.0032). Following georeferenced coordinates, two communities were compared for FGS infection and proximity to lake. As portrayed in kilometers in Fig 2, Mambonkor village (C) which is located farther from the lake had the lowest level of prevalence, unlike Mambonkor Bord (B) which is closest to the lake (Fig 2 and Table 3).

FGS showed no significant association with age (Logistic regression: $X^2$: 3.3; df: 1; p-value = 0.1291); OR: 1.0323), however, younger females reported the highest number of FGS positive cases. More of the unmarried women (98%) were virgins and were not examined visually. This limited the level of association between FGS and marital status. No association was

**Table 3. FGS prevalence distribution per community.**

| Community | FGS prevalence %(n) |
| --- | --- |
| Dakar Mosgum | 33.3(1) |
| Kippo Carrier Kotoko | 50.0(5) |
| Kippo Carrier Mosgum | 72.7(8) |
| Mambonkor Bord | 66.7(6) |
| Mambonkor Village | 20.0(1) |
| Matta Barrage | 75.0(3) |
| Petit Barrage | 80.0(4) |
| Pilusman | 54.0(6) |

*In Ambassadeur, Dakar Arab and Kippo Younous Islands, no visual diagnosis was carried out for FGS prevalence due to inaccessibility related to cultural aspects and unavailability of target population.

reported between intensity of egg excretion and FGS infection ($\chi^2$: 0.9543; df: 1; p-value = 0.3286; OR: 1.0044). Further analysis revealed that FGS only and participants who reported both FGS and eggs in their urine was 40.5%. Females between the ages of 19–23 and 24–28 years reported the highest category of FGS + eggs, while FGS was shown to increase with age. Only married females reported all three categories of infection as shown in Fig 3.

## B. FGS signs and symptoms: Personal symptomatic reports and visual examination

Report of symptoms were gotten through questioning, and physiological signs were recorded by visual examination of the vaginal walls and cervix through colposcopy. Signs and Symptoms

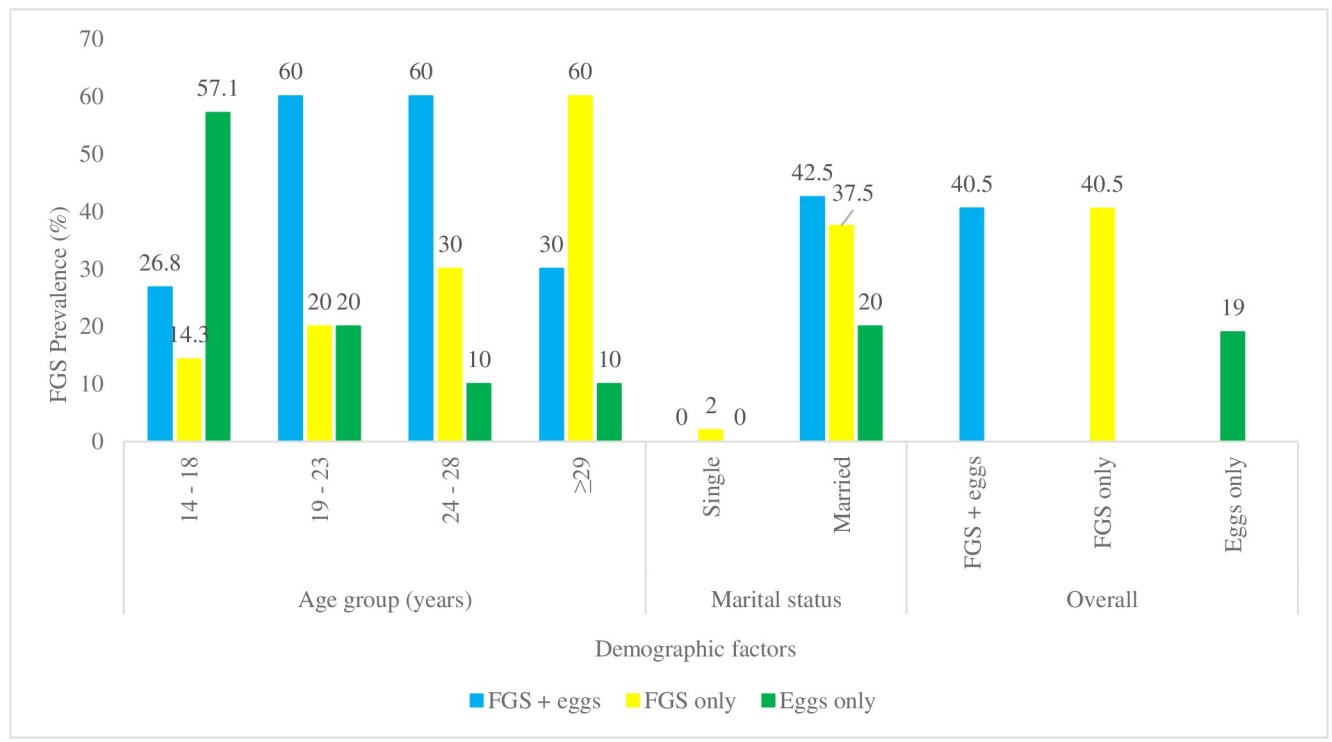

**Fig 3. Categories of infection with respect to age and marital status.**

**Table 4. Summary of self-reported symptoms with confirmation by visual examination by colposcopy.**

| B | A | Visual examination | |
| --- | --- | --- | --- |
| | Sandy patches on homogenous Yellow area+ | Grainy sandy patches+ | Abnormal blood vessels/Vascular changes+ |
| **Reported symptoms (Number of individuals)** | | | |
| Pelvic pain or pain during or after intercourse (32) | 36.2% (21/58) | 36.2% (21/58) | 37.9% (22/58) |
| bleeding or discharge after intercourse (29) | 36.2% (21/58) | 37.9% (22/58) | 34.5% (20/58) |
| vaginal itches (25) | 36.2% (21/58) | 37.9% (22/58) | 34.5% (20/58) |
| smelly vaginal discharge (19) | 24.1% (14/58) | 25.9% (15/58) | 20.7% (12/58) |
| lower abdominal pain (46) | 51.7% (30/58) | 53.5% (31/58) | 50% (29/58) |
| lower back pain (40) | 46.5% (27/58) | 48.3% (28/58) | 44.8% (26/58) |
| Stress incontinence/Urge incontinence (11) | 17.2% (10/58) | 17.2% (10/58) | 17.2% (10/58) |
| Genital itch or burning sensation (32) | 37.9% (22/58) | 36.2% (21/58) | 37.9% (22/58) |
| Inconsistent menstruation/painful menstruation/inter-menstrual bleeding (34) | 39.6% (23/58) | 39.6% (23/58) | 37.9% (22/58) |

recorded from self-reported complaints and self-descriptions from both girls and women were summarized to symptoms recurrent in descriptions: lower abdominal pain, lower back pain, inconsistent/painful menstruation, genital itch or burning sensation, and pelvic pain or pain during intercourse (Table 4). Most commonly identified pathologies from visual inspection through colposcopy were grainy sandy patches, sandy patches on homogenous yellow area, and abnormal blood vessels/vascular changes (Table 4 and Fig 4). There were extremely few cases of cervical polyps (2/34), nabothian cysts (3/34) and rubbery papules (4/34). Of women and girls positive for FGS after both urine analysis and visual examination, menstrual irregularities were common (25/34).

○ **Sub-fertility and FGS**

Infertility was considered as a physical effect of FGS. Although infertility was not very common as a whole with no particular significant association to FGS (Fischer' exact, p-value = 0.1181),

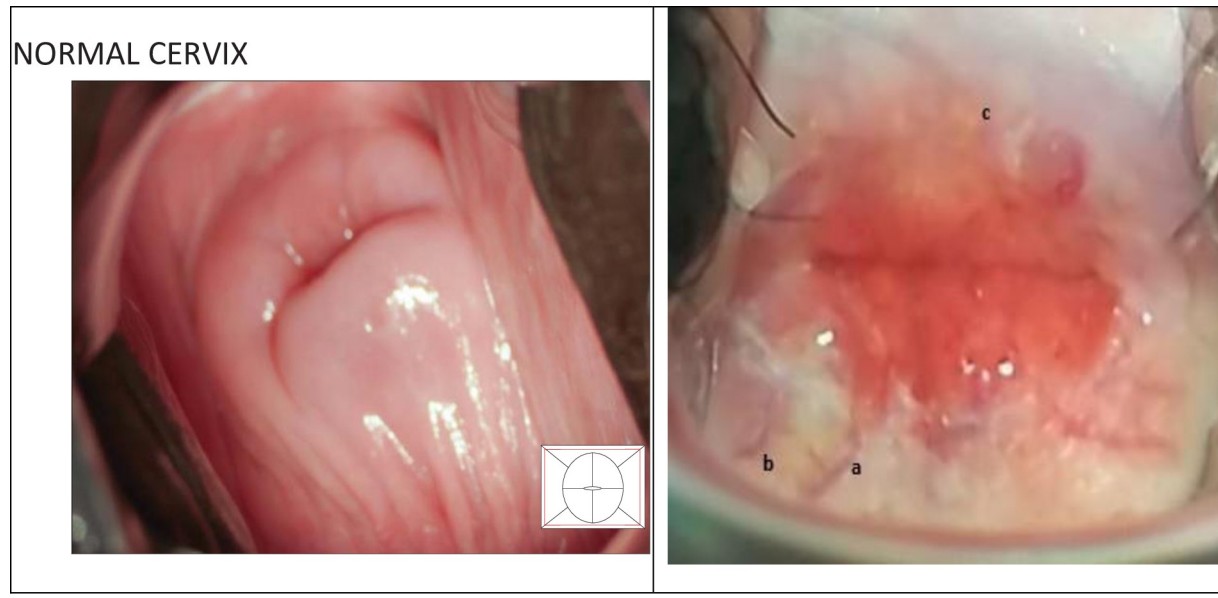

**Fig 4. Picture of healthy cervix and picture of cervix with reported cervical lesions. A.** Normal Cervix (Picture extracted from WHO FGS Pocket Atlas); **B.** Classic FGS scared cervix showing **a)** dilated surface blood vessels and **b)** old well-formed yellowish fibriotic lession, **c)** sandy patches at 12 'Oclock (49 year old woman, not shedding eggs in urine).

with 61.2% of women positive for FGS reported having children, sub-fertility was evident with a significant association found (Fischer' exact, p-value = 0.0049). It was found in our study context, culturally a woman was expected to have children after every 18 months at most, with the number of children a woman births a consideration for her social capital, sub-fertility was rated as a maximum of two years without childbirth in marriage for women of reproductive age. 66.7% of these women positive for FGS had a last child who was at least three (3) years of age ($\chi^2$ = 4.78; p = 0.029), and reported having difficulties in conceiving, because they '*had the worm problem"* or *"maybe had reached menopause"* (for even women in their late 20s within this group).

## C. Socio-economic variables

Amongst physical effects like pain, other effects like stigma, low self-esteem or defeat, and out of pocket expenditure were seen as effects of some symptoms of FGS.

Though the name FGS was new to these women, the overt disease and some of its symptoms were well known with long-suffering effects already considered and felt. Symptoms such as abdominal pain and vagina itches were readily acknowledged and linked to the worms in the water. For vaginal itches, it was mentioned that though several remedies had been sought for treatment, these were unsuccessful and women mentioned "*just living with i*t" or after using both modern treatment and traditional herbs to cut costs, "[but] *it does not solve the itch, it keeps coming back*" (IDIs with FGS positive women after testing).

Young women especially felt a sense of defeat, which affected their self-esteem, from living with the recurrent symptoms for a while. Still regarding vaginal itches, when probed, responses were:

> *"I have done nothing about it, what's there to do? I do know nothing. . .I think it's something normal for women at times? Especially if you have your menses already. . .I just itch* [myself] *until it swells and when I move it is so painful. . ." (IDI with FGS positive woman, late-20s, Pilusman Island)*

> *"The vagina itch keeps coming, even after we use the traditional herbs. . .it is what we know; but it does not solve this itch, it keeps coming back" (IDI with FGS positive woman, mid-40s, Kippo Carrier Kotoko)*

Others regarded this symptom as normal, accepting it as part of their lives as women, because they had lived consistently with it for long, a condition which no treatment or relief had been gotten

> *"I think it is normal for women. . .especially if you are pregnant. . .they say pregnancy brings out the disease in your vagina and there is nothing really to do"(FGD with young women, Pilusman Island)*

> *"I think it is the family planning I took that gives me the vagina itches. . . just like it makes me fat? And the burning urine too" (FGD with young women, Pilusman Island)*

Possible causes of sub-fertility and menstrual irregularities were linked to sexually transmitted infections, which were gotten from husbands through sexual contact with co-wives, causing conflict at home.

> *"She (her co-wife) had it before she came here. . .she gave it to me through my husband, since five years now [I have] no baby, and the pain with my period, even the itches. . . I did not have*

*this before. I suffer because of her"* (IDI with FGS positive woman, mid-30s, Kippo Carrier Mosgum)

○ **Economic cost (Health spending/Cost of treatment)**

Women suffering from FGS reported extra expenditures for health such as direct costs of treatment made up of repeated transportation to point of health care to consult for FGS related symptoms and follow up, consultation fees, and medication.

**i. Out-of-pocket expenditures.** The direct costs of treatment (mainly long term or even significant short term expenditures) from consultation fees, medication, transport, food, assistance, accommodation, were reported as "*very expensive*" or "*unaffordable*". A woman infected with FGS when interviewed explained, up to 300,000 FCFA /500 USD (from her monthly fish revenue and savings) was used in a month for her medical consultations and purchase of drugs.

> « *I have been to the Baptist hospital in Foumban amongst the other ones [health centers visited] for these my problems (vagina itches, discharge, and inability to conceive after first child since moving to Island 09 years ago). . .I spent 300,000 [on medicines and tests] and they did not even tell me what they saw. After all the tests they made me do and the medicines i bought did not offer me any relieve. . .I have left it and will not waste any more money or my time going to the hospital for this again, I will bear, times are hard* » (*female, late-20s, FGS positive, Pilusman Island*)

This was seen to inflict further economic difficulties on a population already struggling to live with less than 500 FCFA (less than 1USD) a day or a total of 50,000 FCFA a month (as per cost of a basket of fish sold on market day). See Table 5.

We measured transportation cost incurred during treatment, a clear picture of the distance and cost of transportation covered to a health center (either of the 3 main hospitals most frequently used)—the health area center, the government district hospital at Malanteoun, and the government regional hospital at Foumban. These hospitals were preferred due to their 'more equipped material and trained medical personnel' who could attend better to these FGS symptom related problems, especially after several tries at the local health center. Table 6 was drawn up to portray cost for treatment in time and money with information collected from participants and from our ethnographical experience.

No doubt, in such instances, it could be justly considered when women shunned going to consult for similar conditions when the experiences of their mates had not yielded any solutions but only more expenditure. Alternatively, these women considered the little they had in hand and preferred persevering, using this on their families or household expenditures rather than invest in their health. This reluctance (and delay) towards diagnosis constructs a path leading to more incurred disabilities and increased suffering, further affecting their economic performance, causing a vicious cycle of poverty and ill health.

**Table 5. Revenue table for fish sale (with 1USD at 550 FCFA).**

| Item | Time used/No of persons | Unit cost per basket | Baskets per Month | Monthly revenue | No of persons covered | Estimate monthly income per person |
|---|---|---|---|---|---|---|
| 1 basket of dried fish | 1 week /3 persons at least | 50,000FCFA/ 90USD | 1–2 | 50,000FCFA/90 USD | Polygamous households with at least 6 people (mostly women and young children) | 8,400FCFA/ 15 USD (averagely 280 FCFA a day) |

**Table 6. Transportation cost table as per geographical location of Island.**

| Distance/place | Cost round trip | Time spent | Result | Observation |
|---|---|---|---|---|
| *Matta health center | By personal fishing canoe free/ commercial **2000FCFA/3.5USD** for round trip (noted here less than 15 percent of population own a canoe, lesser than 2 percent owned by women)<br><br>(or wait and accompany any community member crossing- waste of more time | 3 hours to cross lake (time spent at hospital for consultation not counted here but should be noted exists) | No diagnosis for schistosomiasis (microscopy or urine)<br><br>No diagnosis for FGS (urine or visual)<br><br>No Praziquantel available at hospital pharmacy | No laboratory technician, available microscope, no training on schistosomiasis diagnosis, no knowledge on FGS, no colposcope, awareness of cervical cancer and training on it-FGS could be integrated as recommendation |
| Magba Medical Centre (CMA) | **3000FCFA/5.5USD** from Matta to Magba | 3 hours to Mainland (Matta), and 45 minutes to Magba (up to 2 hours if it rained) | No diagnosis for schistosomiasis (microscopy or urine)<br><br>No diagnosis for FGS (urine or visual)<br><br>No Praziquantel available at hospital pharmacy | No diagnosis for schistosomiasis, personnel trained in diagnosis but no available material for |
| Malanteoun District hospital | **5000FCFA/9USD from Magba or 7000FCFA/12.7USD** directly from Matta | 3 hours to Mainland (Matta), and 45 minutes to Magba (up to 2 hours if it rained), and 3 hours to Malanteoun making a total of 6–7 hours round trip | No diagnosis for schistosomiasis (microscopy or urine)<br><br>No diagnosis for FGS (urine or visual)<br><br>Praziquantel available but must be sent out for distribution only during national MDAs | Boxes of expired Praziquantel in hospital storage waiting to be burned or returned to national program after waiting in vain for authorization to be distributed after several requests from COC |
| Or Foumban regional hospital which was another option sited | 3000FCFA/5.5USD from Matta to Magba<br><br>3000FCFA/505USD from Magba to Foumban<br><br>**Total = 6000FCFA/11USD** | 3 hours to Mainland (Matta), and 45 minutes to Magba (up to 2 hours if it rained), and 3 hours to Foumban district hospital | No diagnosis for schistosomiasis (microscopy or urine)<br><br>No diagnosis for FGS (urine or visual)<br><br>No Praziquantel | This is out of the Malanteoun Health district, but was mentioned in interviews as a well-frequented hospital. This could be due to accessibility because of the poor or at times inaccessible roads and long distance to the district hospital at Malanteoun, people preferred going to Foumban. |

*The Matta health area center does not carry out diagnosis for schistosomiasis though this is an endemic area for urogenital schistosomiasis. Diagnosis and prevalence activities are done only during the National Deworming Program activities by central program staff and health personnel from the central level in Yaoundé. These specifically are during Mass Drug Administration (MDA) campaigns or routine prevalence checks. No individual diagnosis or treatment for schistosomiasis is carried out, considering after MDAs, drugs (Praziquantel) are sent back to the health district as part of the National Program drug control and management policies.

 **ii. Money lost from inability to work.**   Noticeable is not only the direct financial health spending related to treatment seeking from medical institutions, also time lost and indirect labor costs from decreased working hours and reduced economic activity as an effect of morbidity. Here, case study sites were fishing communities or fishing camps, where an individual's revenue is dependent on the amount of fish brought in a day. This relied on factors such as; amount of time spent in placing fish traps (mostly done by men, though in some communities with fewer men, the women filled the gap), and the amount of time spent in the water to bring in the fishes caught and 'fixing' the fish for drying. This later task considered the responsibility of the woman and her children, is time consuming (related to amount of fish brought in). Up to a minimum of 3 hours after the return from fishing is used up for this activity alone, leaving out the time spent catching fish and preparing the home meal. This gives a mediocre idea of the time spent a day for productive and reproductive work, throwing clarity on sick time or treatment seeking time spent out of the island in quest for "*futile' [*as was mostly described] *help for her symptoms*". The transport process as shown in Table 6, takes a minimum of 6 hours and a paid round trip of 3500 FCFA/6USD minimum.

**Table 7. Daily productive activity table to portray cost of disease for girls ≥ 13 and women (based on an average income of 280 FCFA/day (from Table 4)).**

| Activity | Hours spent in productive work or economic activity/day | work hours lost a day | Amount lost per activity (at 280frs made a day)–FCFA | Group involved | |
|---|---|---|---|---|---|
| | | | | Girls ≥ 13 | Women ≥18 |
| Fishing Net and basket weaving | 2 | 2 | 44 | | X |
| Trap setting | 2 | 2 | 44 | | X |
| Checking traps (fishing nets set) | 1 | 1 | 22 | X | X |
| Beating water to attract fish (Tappé) | 1 | 1 | 22 | | X |
| Fishing (collecting fish from traps or net set) | 2 | 2 | 44 | | X |
| Fixing of fish and preparing for drying | 1 | 1 | 22 | X | X |
| Drying of fish | 2 | 2 | 44 | X | X |
| Packaging of dried fish | 0.5 | 0.5 | 11 | X | X |
| Farming | 1.05 | 1.05 | 33 | X | X |
| Sale on market day (1 main market day per week per Island, travelling to market site and back and time used to sell fish = averagely 10 hours. | / | / | / | | X |
| Total | **13** | **13** | **286FCFA/0.52USD** | | |

We set out to calculate the amount of time lost and cost of disease per day, based on an average working week of six (6) days, where Sunday was a free day for social networking, and Friday market day. In Table 7 below, listed are the minimum activities carried out by the girl/woman a day, with the accrued time used, time lost to sickness, and related cost. Friday, reserved for marketing of fish as mentioned above, it is listed here but not included in our calculations (this was a day in the week which was reserved for the single activity of going to the market).

**iii. Opportunity cost.** Women's role at the home as defined by culture and other variables, were quantified (opportunity cost) in an attempt to measure the effect and cost of FGS (Table 8). We looked at a measure of the different household activities or responsibilities (reproductive labour) [48] of the girl/woman in the family, and how much this could be translated to if quantified a day. Apart from emotional support which could not be quantified, an estimated cost and hours lost a day could be seen as a result of the disease. This is to portray how constraints from symptoms and pathology from FGS affects the girl/woman and her entourage directly.

## Discussion

### A. Socio-demographic variables and effect on FGS infection

**Age and infection.** Amongst variables observed in this study, the association between age and FGS infection was almost negligible ($\chi^2$: 3.3; df: 1; p-value = 0.1291; OR: 1.0323), but a

**Table 8. Opportunity cost of disease.**

| 'Second duty' or home chores unquantified or not costed (Opportunity Cost) | Period | Time used per day (hours) | Translated Cost per day (at 280frs gained in a day of 12 working hours) -FCFA |
|---|---|---|---|
| Cooking | Morning and/evening | 2 | 44 |
| Washing dishes, clothes and other household chores (household labor) | Morning period from 5–7 or 11 to 2pm after returning from fishing | 2 | 44 |
| Cleaning and feeding of baby and family (nurturing) | Full day when even when carrying out other chores around home | 3 | 66 |
| social networking for family | Most evenings (and all of Sunday) | 2 | 44 |
| Emotional support | Full day (except when not at sleep) | / | / |
| **Total hours of opportunity cost a day (except for emotional support which was not quantified)** | / | **9 hours/day** | **198 FCFA/day** |

slight spike in infection with FGS was found amongst young women in their 20s to 30s which is consistent with other studies [49]. Higher infection amongst this age group has been reported as reason of younger women having more acute infections, compared to older women who have more severe chronic disease from life-time duration of chronic sequelae of infection [7, 50]. This group (20s to 30s) was found to be made up of mostly young married women, actively involved in fishing and household chores for the family, thus more water contact, and with a history of these activities long before, as was reported by respondents when questioned of their activities. This emphasizes the current and future risk of young girls and young women to infection with FGS due to their gender roles such as- for young wives, whose children were still young and they had to do all the chores; or unmarried girls who carried out these chores for their mothers and household. *"It's her who washes clothes. . .her children are still young"*, was a common response received within this study when respondents were questioned about activities of this age group. This draws attention to the susceptibility of these young women and girls to infection with FGS [45], intensifying the need for more preventive and treatment focus of this group [51] and especially in childhood [25], with the recent advancement in developing of pediatric Praziquantel [52]. Related, menstrual irregularities (painful, irregular or seized menstruation found to be prevalent amongst 73.5% of FGS positive females), amongst post menarche girls with UGS, added to symptoms like vaginal itches and burning sensation in genitals, already asserted to infer FGS in young girls [15, 53], was identified as signify possible presence of FGS amongst this group, especially in such resource limited settings.

**Culture and infection.** Among individuals with FGS results, the reported activities were farming, fishing and farming/fishing. We looked at the values first singly. Looking solely at the economic culture, we found no significant association between FGS and economic activity practiced (0.5005; df: 2; p-value: 0.7785). After examining these variables singly, we used interrelations such as proximity to lake and socioeconomic culture, which are connected for a deeper analysis. Proximity to lake bred a culture of fishing and dependence on lake for household water source and creation of wells [33]. A significant association was found between FGS infection and these combined variables, when comparing two Islands with differing lake proximity and culture (example of Mambonkor Village and Mambonkor Bord). Mambonkor village (unlike Mambonkor Bord with fishing culture; >64% FGS infection rate) is farthest from the lake ($\geq$ 1.085km) and consists mostly of the Tikar ethnic group, who are culturally farmers. This community had the least cases of FGS (1/34, <20%), and it was noted from interviews with respondents, infected persons found here reported to have originally lived in Mambonkor Bord (closest community to lake- 0.5 km), before relocating due to marital relations or family ties. This showed a pattern that older women though having less/no eggs, possibly suffer chronic effect of infection they aqcuired in childhood [25], leaving a lasting signature, pointing to the importance for diagnosis.

## B. Socio-economic and psychosocial cost of FGS

The delayed diagnosis of disease and progression over periods of years and decades cause greater and largely irreversible physical impairment as well as huge economic and social consequences from seeking frequent treatment sources from within and out of the health system [54]. These effects of diseases, especially NTDs, add to the already existing burden—a physical and economic handicap in poor rural settings, where physical labor is the major subsistence means, causing productivity loss such as presentism, absenteeism, and loss of workdays in these mostly agricultural societies [38, 55]. Similarly, in the lives of women, FGS was seen to cause increased out-of-pocket expenditure for treatment, and prolonged suffering from late diagnosis.

**Women's health and capital.** The peculiar nature of women's responsibilities both in economic production and within the family (reproductive labour [48]), has been argued to have a profound impact on the extent to which they are affected by diseases and their response to these diseases [56]. Both biomedical and social research on effects of neglected tropical diseases on women has been primarily focused mostly on the gender differences related to pregnancy and reproduction, with broader social roles and responsibilities minimized [48, 56]. The cost of illness is not only material or financial, but a heavy sufferance both social and mentally (low self-esteem, despair) is felt on the infected and affected [57, 58].The loss of social capital, which are social resources from the individual or group that can be gained and used to reach individual or collective goals [59, 60], was seen as a huge effect of FGS in the lives of women. The woman's role and ability to exert power within the group, who through her income generating activities or housekeeping, contributes a huge bulk to the economy of the family, was threatened due to infection with FGS. Her ill health from the disease affects herself and the family or group's cultural capital directly. This cultural capital, through institutionalized forms such as the body, the mind, marriage and motherhood, are challenged, where the woman's success in marriage, child birthing; and also, the satisfaction (sexual) given to a husband are weakened through her illness. Suffering from FGS, this cultural value of the woman is attacked, directly affecting all aspects of her life, such as her social ties causing poor mental health in the form of self-stigma or external stigma [26].

Infertility /sub-fertility has been considered as a physical effect of FGS [10, 16, 24, 27]. Marriage and children are a symbolic capital [59], defining legitimate competence and authority within our case study societies, and if not met, are a shortcoming or a reduction of her natural or cultural and social capacity within her entourage. Out of the women infected with FGS, 66.7% of these women had a last child who was at least three years of age, and reported having difficulties in having more children. Similarly, Beekhuizen and Sharma [61] reiterate our findings with sub-fertility existing when a couple has "not achieved pregnancy after 2 years of having regular unprotected sex". The magnitude of this 'mostly neglected' and 'low priority' public health problem, which prevails amongst 13% and 32% of the population in sub-Saharan Africa has been reported, with social stigma and exclusion from community an effect of infertility or sub-fertility [61].

## C. Access to diagnostics and treatment

Concurring with current views [22] on the need for diagnostic access and rapidity for FGS which touches keenly physiologically and socio-economically the infected person, the lack of diagnostics and treatment is a major cause of suffering amongst women and girls with FGS. Delayed FGS diagnosis directly leads to a cycle of pain and impoverishes the infected women because of prolonged expenditures on futile diagnosis and treatments for STIs and other suspected infections. Research for diagnosis on genital schistosomiasis is on the uprise [13, 14, 62], with most recent research [49] showing a more adaptable diagnostic tool for research poor settings. Recently, an FGS intervention manual for health workers was developed [63], and training workshops were held (March 2020, May 2021) [64] where over 100 health workers from sub-Saharan African countries received training on FGS prevention, treatment and diagnosis. Still, there is more need for further research around diagnosis, treatment and control, especially in endemic communities where this is almost non-existent. This request does not seem too farfetched especially if existing drug procurement and distribution paths for schistosomiasis already exist, which can be exploited further to avail treatment at health centers and with health workers in endemic communities and also more frequent mass drug administrations [39].

This study did not fully consider a representation of males (except from community drug distributors, a few husbands who chose to speak for their wives, and male health workers) to understand their own perceptions concerning most aspects considered here, except for aspects of productivity costs and activities which enriched our data. Notwithstanding, this was considered in depth from women who are our main participants, and whose experiences we sought to acquire. In addition, like in most other studies on FGS diagnosis, the diagnosis of FGS in young girls pre-menarche and/or virgins was limited to questioning which led only to inferred conclusions.

## Conclusion

Our study reports infection with FGS (58.6%) associated with lake proximity and socio-cultural aspects amongst girls and women in the Matta health area in Cameroon. Lived experiences of FGS infected women/girls showed psychosocial effects, productivity loss and catastrophic out of pocket expenditures of up to 500USD for treatment seeking for related gynecological illness as a result of infection. Looking at these daunting socio-economic and physiological challenges lived and experienced by these women, indeed, it is time that « local realities are put at the forefront of intervention planning » [65], in incorporating the voices, opinions, experiences and capacities of infected persons in the control of diseases. In addition to the remarkable scale-up with schistosomiasis mapping and integration of most aspects of the PHASE approach in Cameroon, effective actions of health personnel both at the primary health care level as the fore-fighters and those at the central level as the decision makers is still needed to successfully prevent, treat and support FGS.

## Supporting information

**S1 Fig. Matta health area data from Malanteoun health district census, 2018–2020.** (TIF)

**S1 Text. Close-ended structured questionnaire and in-depth interview/FGD guides.** (DOCX)

## Acknowledgments

We gratefully acknowledge the moral and intellectual support from the "Projet MTN/ OCEAC" team, with much gratitude to the Regional Coordinator, Dr. Savadogo Bonaventure, for guidance and overall supervision. Also, our gratitude to the District Medical Officer of the Malanteoun Health District, Dr. Amabo Elvis, for his support in this study, and all the staff and informal health workers (CHWs, CDDs, traditional birth attendants) of the Matta Barrage Health Area for their kind support and invaluable assistance. We thank all the participants of this study for according us their time and trust, and for their courage in sharing the samples and personal experiences with us. Thanks to the women and girls of the fishing camps/Islands surrounding Matta Barrage for trusting us with sharing their stories.

## Author Contributions

**Conceptualization:** Makia Christine Masong.

**Formal analysis:** Makia Christine Masong, Godlove Bunda Wepnje, Victoria Gamba, J. Russell Stothard.

**Funding acquisition:** Makia Christine Masong.

**Investigation:** Makia Christine Masong, Godlove Bunda Wepnje, Ntsinda Tchoffo Marlene.

**Methodology:** Makia Christine Masong, Godlove Bunda Wepnje, Marie-Therese Mengue, Estelle Kouokam, J. Russell Stothard, Albert Legrand Same Ekobo.

**Project administration:** Makia Christine Masong.

**Supervision:** Marie-Therese Mengue, Estelle Kouokam, J. Russell Stothard, Albert Legrand Same Ekobo.

**Writing – original draft:** Makia Christine Masong.

**Writing – review & editing:** Makia Christine Masong, Godlove Bunda Wepnje, Ntsinda Tchoffo Marlene, Victoria Gamba, Marie-Therese Mengue, Estelle Kouokam, J. Russell Stothard, Albert Legrand Same Ekobo.

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
