## [Decision Letter · Decision Letter 0]

26 Jul 2021

 PGPH-D-21-00270 Female Genital Schistosomiasis (FGS) in Cameroon: A formative epidemiological and socioeconomic investigation in eleven rural fishing communities. PLOS Global Public Health

Dear Dr. Masong,

Thank you for submitting your manuscript to PLOS Global Public Health. After careful consideration, we feel that it has merit but does not fully meet PLOS Global Public Health’s publication criteria as it currently stands. Therefore, we invite you to submit a revised version of the manuscript that addresses the points raised during the review process.   Please submit your revised manuscript by Sep 09 2021 11:59PM. If you will need more time than this to complete your revisions, please reply to this message or contact the journal office at globalpubhealth@plos.org. Please include the following items when submitting your revised manuscript: A rebuttal letter that responds to each point raised by the editor and reviewer(s). You should upload this letter as a separate file labeled 'Response to Reviewers'.A marked-up copy of your manuscript that highlights changes made to the original version. You should upload this as a separate file labeled 'Revised Manuscript with Track Changes'.An unmarked version of your revised paper without tracked changes. You should upload this as a separate file labeled 'Manuscript'.  

We look forward to receiving your revised manuscript.

Kind regards,

Hannah Tappis, DrPH, MPH

Academic Editor

Journal Requirements:

Additional Editor Comments (if provided):

This is a well designed and reported study on an under-addressed topic in global public health. Both reviewers have provided constructive suggestions for minor revisions to strengthen the publication.

Reviewers' comments:

Reviewer's Responses to Questions

**Comments to the Author**

1. Does this manuscript meet PLOS Global Public Health’s publication criteria? Is the manuscript technically sound, and do the data support the conclusions? The manuscript must describe methodologically and ethically rigorous research with conclusions that are appropriately drawn based on the data presented.

Reviewer #1: Yes

Reviewer #2: Yes

2. Has the statistical analysis been performed appropriately and rigorously?

Reviewer #1: Yes

Reviewer #2: Yes

3. Have the authors made all data underlying the findings in their manuscript fully available (please refer to the Data Availability Statement at the start of the manuscript PDF file)?

Reviewer #1: Yes

Reviewer #2: Yes

4. Is the manuscript presented in an intelligible fashion and written in standard English?

Reviewer #1: Yes

Reviewer #2: Yes

5. Review Comments to the Author

Reviewer #1: This is an interesting, thorough and important article, that uses both quantitative and qualitative methods to show the impact of local health problems on livelihoods in poor and remote settings. Beyond the documentation of how important a globally neglected problem such as FGS can be at the local level, it highlights what global rhetoric like Universal Health Coverage and Health Systems Strengthening really means in a local setting. And the road to achieving these goals still appears to be long.

I just have the following suggestions for further improvement of the manuscript:

- On line 88, I suggest adding the following paper to the cited references - Patel P, Rose CE, Kjetland EF, Downs JA, Mbabazi PS, Sabin K, Chege W, Watts DH, Secor WE. Association of schistosomiasis and HIV infections: A systematic review and meta-analysis. Int J Infect Dis. 2021 Jan;102:544-553. doi: 10.1016/j.ijid.2020.10.088. Epub 2020 Nov 3. PMID: 33157296. This paper is recent and provides the most compelling evidence of an association between SCH and HIV.

- Remove the repetition in lines 369 thru 371

- Please add Figure 5 – I could not see it in the pdf file that I was asked to review

- Please check (and correct, if necessary) the following references regarding evidence for underpinning the statements made in the body of text – refs 7, 8, 42, 43.

Furthermore, there are some editorial corrections to be made, but I will leave this to the editor.

Without hesitation, I recommend this manuscript for publication in PLOS Global Public Health with the suggested minor changes

Reviewer #2: This is a good manuscript which describes important research findings on a less- studied aspect of a prevalent NTD, schistosomiasis, causing FGS, debilitating consequences to women in poor, rural endemic areas of Sub – Saharan Africa.

My comments include the following:

Line 97–8: Describe more clearly the ‘main objective of the study’, as this is very broad and ambiguous

Line 244: Include word ‘Colposcope’ after words ‘hand-held’

Line 326-30: I suggest move these lines to after line 342. Section A. should start with i. Demographic characteristics…

Line 329: Should add ‘any’ between ‘presence of’ and ‘three main parameters’

Line 334-5: Rephrase the last part of the sentence, can read ‘most of them (88.2%) living in a proximity of <100m’

Line 335-42: Start with heading ii. Parasitological and clinical description…, and then have all lines under this.

Line 339: Remove word ‘both’

Line 348: Table 2 should move to the ‘Demographic characteristics…’, after line 335

Line 531: Remove the statistical values ‘Logistic regression’ since presented above in the results

Line 532-5: Another explanation for higher infection in 20s to 30s would be their likely to have more acute infections compared to older women who will be having severe chronic disease. You may need to include this.

Line 548: Remove the statistical values ‘ Chi-squared’

Figure 3: Consider removing the error bars, they do not add more value to the already interesting results presented in the figure.

Figure 4: Excellent, very important figure for this paper

6. PLOS authors have the option to publish the peer review history of their article (what does this mean?). If published, this will include your full peer review and any attached files.

**Do you want your identity to be public for this peer review?** For information about this choice, including consent withdrawal, please see our Privacy Policy.

Reviewer #1: No

Reviewer #2: No

---

## [Editor Report · Decision Letter 1]

15 Sep 2021

Female Genital Schistosomiasis (FGS) in Cameroon: A formative epidemiological and socioeconomic investigation in eleven rural fishing communities.

PGPH-D-21-00270R1

Dear Dr. Masong,

We're pleased to inform you that your manuscript has been judged scientifically suitable for publication and will be formally accepted for publication once it meets all outstanding technical requirements.

Within one week, you'll receive an e-mail detailing the required amendments. When these have been addressed, you'll receive a formal acceptance letter and your manuscript will be scheduled for publication.

An invoice for payment will follow shortly after the formal acceptance. To ensure an efficient process, please log into Editorial Manager at https://www.editorialmanager.com/pgph/ click the 'Update My Information' link at the top of the page, and double check that your user information is up-to-date. If you have any billing related questions, please contact our Author Billing department directly at authorbilling@plos.org.

Kind regards,

Hannah Tappis, DrPH, MPH

Academic Editor
